# BEHAVIORAL EMBEDDINGS OF PROGRAMS: A QUASI-DYNAMIC APPROACH FOR OPTIMIZATION PREDICTION

**Haolin Pan**[1,2,3]**, Jinyuan Dong**[1,2]**,Hongbin Zhang**[2]**,Hongyu Lin**[1,2]**, Mingjie Xing**[1,2,3][*]**Yanjun Wu**[1,2]

[1]University of Chinese Academy of Sciences, China

[2]Institute of Software Chinese Academy of Sciences, China

[3]Hangzhou Institute for Advanced Study at UCAS, China

{panhaolin21,dongjinyuan24}@mails.ucas.ac.cn
{hongbin2019,hongyu2021,mingjie,yanjun}@iscas.ac.cn,

## ABSTRACT

Learning effective numerical representations, or embeddings, of programs is a fundamental prerequisite for applying machine learning to automate and enhance compiler optimization. Prevailing paradigms, however, present a dilemma. Static representations, derived from source code or intermediate representation (IR), are efficient and deterministic but offer limited insight into how a program will behave or evolve under complex code transformations. Conversely, dynamic representations, which rely on runtime profiling, provide profound insights into performance bottlenecks but are often impractical for large-scale tasks due to prohibitive overhead and inherent non-determinism. This paper transcends this trade-off by proposing a novel quasi-dynamic framework for program representation. The core insight is to model a program's optimization sensitivity. We introduce the Program Behavior Spectrum, a new representation generated by probing a program's IR with a diverse set of optimization sequences and quantifying the resulting changes in its static features. To effectively encode this high-dimensional, continuous spectrum, we pioneer a compositional learning approach. Product Quantization is employed to discretize the continuous reaction vectors into structured, compositional sub-words. Subsequently, a multi-task Transformer model, termed PQ-BERT, is pre-trained to learn the deep contextual grammar of these behavioral codes. Comprehensive experiments on two representative compiler optimization tasks—Best Pass Prediction and -Oz Benefit Prediction—demonstrate that our method outperforms strong baselines. Our code is publicly available at [1].

## 1 INTRODUCTION

Applying machine learning to automate and enhance compiler optimization has emerged as a promising direction to unlock the full potential of modern complex hardware Ashouri et al. (2018); Pan et al. (2025a;b); Chen et al. (2021); Ansel et al. (2014); Deng et al. (2025). The success of this paradigm hinges on a fundamental prerequisite: learning an effective numerical representation, or *embedding*, of a program. A powerful program embedding acts as a universal semantic interface, capturing the essential properties of the source

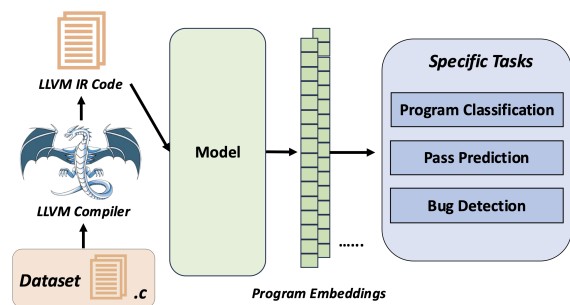

Figure 1: Example of machine learning for compiler optimization task.

---

[*]Corresponding author.

[1]Code: https://github.com/Panhaolin2001/PREP/

code in a dense vector. As illustrated in

Figure 1, such representation can serve as a foundational component for a diverse array of high-impact downstream tasks, ranging from optimization prediction and code classification to bug detection and performance analysis.

However, learning a representation that is both rich in semantics and practical for real-world compilers presents a fundamental dilemma, forcing a choice between two prevailing but flawed paradigms: (1) **Static Representations:** This dominant approach extracts features from various static program representations, including source code, intermediate representation (IR), abstract syntax trees (ASTs), and control or data flow graphs (CFGs/DFGs) Wei et al. (2020); Hellendoorn et al. (2019); Guo et al. (2020). Methods range from handcrafted feature vectors like Autophase Haj-Ali et al. (2020b) to deep learning models that operate on sequences (e.g., IR2Vec VenkataKeerthy et al. (2020)) or graph structures Cummins et al. (2021a); Guo et al. (2020). The primary advantage of static methods is their efficiency and determinism. However, their core limitation is their myopic nature: they describe what a program *is*, structurally, but offer limited insight into how it will *behave* or *evolve* under complex code transformations. (2) **Dynamic Representations:** An alternative approach involves profiling the program during execution to collect runtime features, such as hardware performance counters (HPCs) Xu et al. (2023). These representations offer profound insights into a program's true performance bottlenecks. Nevertheless, they are often impractical for large-scale tasks due to their prohibitive overhead and inherent non-determinism. This trade-off between the efficiency of static analysis and the insightfulness of dynamic profiling has created a significant bottleneck, limiting the capabilities of current learning-based compilers.

In this work, we transcend this dilemma by proposing a novel quasi-dynamic framework for program representation. Our core insight is that an effective representation for optimization can model a program's optimization sensitivity—its intrinsic propensity to react to different code transformations. We introduce the **Program Behavior Spectrum**, a new representation generated by *probing* the program's IR with a set of diverse optimization sequences and quantifying the resulting changes in its static features. To ensure scale-invariance, these reactions are captured using a logarithmic relative difference. We then pioneer a compositional learning approach to encode this high-dimensional spectrum: **Product Quantization (PQ)** discretizes the continuous reaction vectors into structured *sub-words*, and a multi-task Transformer model (`PQ-BERT`) is pre-trained to learn their deep contextual grammar.

Our main contributions are threefold:

- We are the first to propose a *quasi-dynamic* program representation, the Behavioral Spectrum. It captures a program's optimization sensitivity by measuring changes in static features under carefully designed optimization probes.

- We present a compositional encoding methodology using Product Quantization (PQ) and a tailored multi-task Transformer (`PQ-BERT`). This combination effectively learns the deep grammar of program behavior while addressing the scale-vs-precision trade-off in representation learning.

- We introduce a program embedding specifically designed for compiler optimization, and demonstrate through comprehensive experiments that our method, `Behavioral-PQ`, outperforms other baselines on two representative compiler optimization tasks.

## 2 METHODOLOGY

Our proposed framework learns program representations by modeling their *quasi-dynamic* reactions to compiler optimizations. The process consists of three main stages, as illustrated in Figure 2: (1) **Behavioral Spectrum Extraction**, where we quantify a program's optimization sensitivity; (2) **Structured Vocabulary Construction**, where we encode the continuous spectra into discrete, compositional vocabulary; and (3) **Behavioral Grammar Learning**, where a Transformer model learns the deep contextual relationships within these vocabulary.

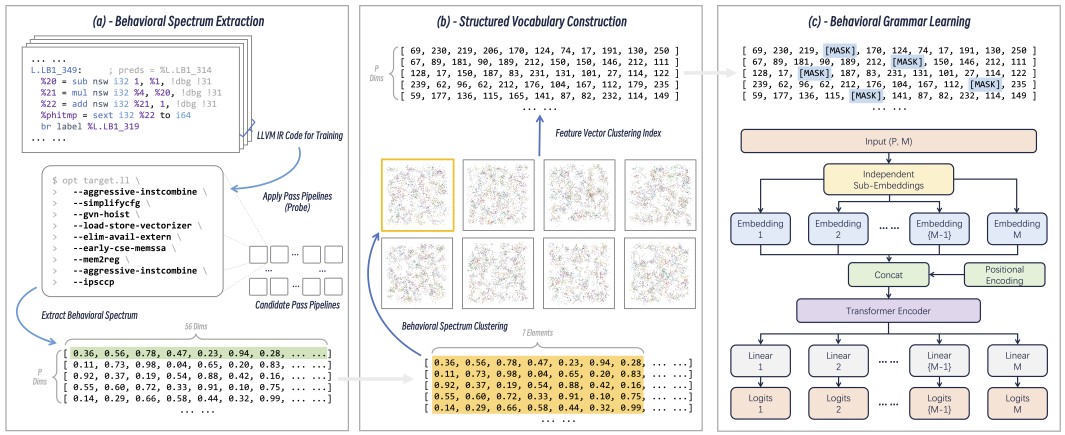

Figure 2: The overall architecture of our program representation learning framework. It transforms LLVM IR into a Behavioral Spectrum, encodes it using Product Quantization, and learns its underlying grammar via a pre-trained Transformer model.

## 2.1 STEP 1: BEHAVIORAL SPECTRUM EXTRACTION

The foundation of our approach is to represent a program not by its static structure alone, but by its Behavioral Spectrum: a high-dimensional footprint that characterizes its reactions to a diverse set of optimization transformations.

### 2.1.1 PROBING PROGRAM BEHAVIOR.

To elicit a program's behavior, we apply a set of carefully designed optimization sequences, termed as probes, to its LLVM IR. The choice of probes is critical: random or single-pass selections cannot reveal the nuanced interactions present in realistic compilation.

We construct probes systematically in a data-driven way. First, each program in the pre-training corpus is represented by its baseline Autophase feature vector, $h_{orig}$. We adopt Autophase because it is a 56-dimensional feature set (e.g., counts of arithmetic instructions, basic blocks, and branches) that provides a stable summary of program structure, and its changes under optimization directly capture behavioral variation. We cluster these representations into $P$ groups, under the assumption that programs with similar structural features exhibit similar optimization sensitivities Haj-Ali et al. (2020b). For each cluster, we employ a heuristic search Garciarena & Santana (2016) (e.g., genetic algorithm or greedy strategy) to find one fixed-length sequence that maximizes the average instruction count reduction across the cluster. Instruction count reduction is chosen as the optimization objective because it is a reliable indicator of code size improvement.

This process yields $P$ distinct sequences, each tuned to a category of program structures. The resulting probe set is both powerful, since each probe is empirically optimized, and diverse, since different clusters give rise to different strategies. For any program $p$, we then compute its behavioral spectrum by first extracting its baseline vector $h_{orig} \in \mathbb{R}^{56}$, and then applying each probe $i \in 1, \ldots, P$ to obtain optimized versions $p'_i$ with corresponding vectors $h_{opt,i}$. Comparing $h_{orig}$ with $h_{opt,i}$ across all probes produces a rich spectrum of behavioral transformations.

### 2.1.2 SCALE-INVARIANT REACTION QUANTIFICATION

A significant challenge in comparing program behaviors is their inherent scale sensitivity. A transformation that removes 100 instructions may be monumental for a small kernel but trivial for a million-line application. To address this, we quantify the reaction not as an absolute difference, but as a **logarithmic relative difference**, ensuring a scale-invariant representation. For each feature dimension $j \in \{1, \ldots, 56\}$, the reaction $d_{i,j}$ for probe $i$ is calculated as:

$$d_{i,j} = \log(1 + \max(0, h_{opt,i,j})) - \log(1 + \max(0, h_{orig,i,j})) \tag{1}$$

where $\max(0, \cdot)$ ensures that the input to the logarithm is non-negative, robustly handling potential minor negative values arising from feature extraction artifacts. The $\log(1 + x)$ transformation (or `log1p`) gracefully handles zero-valued features and compresses the effect of large absolute changes, focusing on multiplicative, order-of-magnitude shifts. The complete Behavioral Spectrum for program $p$ is thus a matrix $\mathbf{S}_p \in \mathbb{R}^{P \times 56}$, where each row is a scale-invariant reaction vector. This corpus of spectra forms the basis for all subsequent learning.

## 2.2 STEP 2: STRUCTURED VOCABULARY CONSTRUCTION VIA PQ

### 2.2.1 MOTIVATION FOR DISCRETIZING BEHAVIORAL SPECTRA

The Behavioral Spectrum, $\mathbf{S}_p$, provides a rich, continuous representation of a program's optimization sensitivities. However, to learn the complex grammar and long-range dependencies within this $P$-step sequence, we need to leverage powerful sequence models like Transformers. These models traditionally operate on discrete tokens, necessitating a bridge from our continuous vector space to a discrete vocabulary.

The motivation for this discretization stems from a core hypothesis in compiler optimization: **generalization**. A given optimization sequence often elicits a similar *type* of behavioral change across a range of different programs. For example, a loop-unrolling pass might consistently produce a *reaction* characterized by an increase in arithmetic instructions and a decrease in control-flow instructions, regardless of the specific program's details. It is therefore desirable to group these similar continuous reaction vectors into a finite set of discrete *behavioral archetypes* or *words*. Clustering is a natural approach for discovering such archetypes.

However, a naive *hard* clustering method, which assigns each vector to a single, indivisible cluster ID, may suffer from **information loss**. A vector lying on the boundary between two clusters is forced into a single choice, losing the valuable information that it shares characteristics with both archetypes. Furthermore, this approach struggles to capture the fine-grained internal structure of the 56-dimensional reaction vectors. To overcome these limitations, we employ **Product Quantization** Jegou et al. (2010), a structured vector quantization technique that performs clustering in a more granular, compositional manner.

### 2.2.2 PQ FOR STRUCTURED BEHAVIORAL ENCODING

The core insight of PQ is to abandon the search for monolithic, high-dimensional prototypes and instead learn a set of low-dimensional, reusable *building blocks* or *primitives*. It posits that any complex reaction vector can be approximately reconstructed by *composing* these simpler primitives. This is analogous to representing a complex color not as a single entry in a giant color palette, but as a combination of fundamental R, G, B values.

To implement this, PQ decomposes each $D = 56$ dimensional reaction vector $\mathbf{d}$ into $M$ disjoint sub-vectors. In our work, we choose $M = 8$, resulting in 8 sub-vectors $\{\mathbf{d}_1, \ldots, \mathbf{d}_8\}$, each of dimension $D_{\text{sub}} = D/M = 7$. A separate, small-scale K-Means quantizer, $q_i$, is then trained for each of the $M$ subspaces Ahmed et al. (2020); Ravuri & Amarasinghe (2025). Each quantizer $q_i$ has its own codebook (or *sub-vocabulary*) $\mathcal{C}_i = \{\mathbf{c}_{i,1}, \ldots, \mathbf{c}_{i,k^*}\}$ containing $k^* = 256$ low-dimensional centroids (where $k^*$ corresponds to $n_{\text{bits}} = 8$).

An arbitrary reaction vector $\mathbf{d}$ is then encoded by quantizing each of its sub-vectors $\mathbf{d}_i$ independently:

$$c_i = q_i(\mathbf{d}_i) = \arg\min_j \|\mathbf{d}_i - \mathbf{c}_{i,j}\|^2 \tag{2}$$

The final representation is a **compositional code**: a tuple of $M = 8$ integer IDs, $\mathbf{c} = (c_1, c_2, \ldots, c_8)$, where $c_i \in \{0, \ldots, 255\}$. The original vector $\mathbf{d}$ can be approximately reconstructed by concatenating the corresponding centroids: $\hat{\mathbf{d}} = [\mathbf{c}_{1,c_1}, \mathbf{c}_{2,c_2}, \ldots, \mathbf{c}_{8,c_8}]$.

This compositional approach allows us to represent a vast number of distinct vectors (a virtual vocabulary of size $256^8$) with a compact set of learned centroids ($8 \times 256$), thereby retaining fine-grained structural information with minimal loss.

## 2.3 STEP 3: LEARNING THE BEHAVIORAL GRAMMAR WITH PQ-BERT

The PQ step transforms each program's continuous Behavior Spectrum into a structured, discrete *article* of size $P \times M$. This encoding preserves fine-grained information but does not yet capture the rich contextual dependencies between the $P$ different reactions. For instance, a program's strong reaction to a loop-unrolling probe is often correlated with its reaction to a vectorization probe. To effectively learn this deep, underlying *grammar* of compiler optimization behavior, we require a sufficiently powerful and expressive sequence model.

We design a Transformer-based model, which we call **PQ-BERT**, tailored to our compositional codes. Standard language models like BERT are trained to predict a single token from its context. However, our PQ representation is multi-faceted, where each reaction is described by $M$ sub-word IDs. A naive approach might concatenate these sub-words into a longer sequence, but this may disrupt the inherent synchrony of the $M$ subspaces. Therefore, our method is to treat the prediction of the $M$ sub-word IDs as a **multi-task learning problem**, where the model is forced to learn the intricate correlations between subspaces simultaneously. We pre-train PQ-BERT using a **multi-task Masked Language Model (MLM)** objective.

**Model Architecture.** The PQ-BERT architecture is designed to process the $P \times M$ matrix of compositional codes. The input to the model is a sequence of $P$ reaction codes, $\mathbf{C} = \{\mathbf{c}_1, \mathbf{c}_2, \ldots, \mathbf{c}_P\}$, where each $\mathbf{c}_t = (c_{t,1}, \ldots, c_{t,M})$ is a tuple of $M$ sub-word IDs. The model first projects these discrete codes into a continuous vector space using $M$ independent sub-embedding layers. For each code $\mathbf{c}_t$, the $i$-th sub-embedding layer, $E_i$, maps the sub-word ID $c_{t,i}$ to a low-dimensional vector $\mathbf{e}_{t,i} = E_i(c_{t,i})$. These $M$ sub-embeddings are then concatenated to form a single high-dimensional embedding $\mathbf{x}_t = [\mathbf{e}_{t,1}; \mathbf{e}_{t,2}; \ldots; \mathbf{e}_{t,M}]$ for the $t$-th reaction, where its dimension is $D_{\text{model}} = 256$. This sequence of $P$ fused embeddings, $\mathbf{X} = \{\mathbf{x}_1, \ldots, \mathbf{x}_P\}$, is then augmented with positional encodings and processed by a standard multi-layer Transformer Encoder Vaswani et al. (2017). The encoder uses self-attention to produce a sequence of contextually-aware output representations $\mathbf{H} = \{\mathbf{h}_1, \ldots, \mathbf{h}_P\}$. Finally, to facilitate the multi-task objective, the model employs $M$ independent linear output heads. The $i$-th head, $O_i$, takes the entire output sequence $\mathbf{H}$ and produces a distribution of logits over the $k^*$ possible sub-words for the $i$-th subspace.

**Pre-training Task and Objective Function.** We pre-train PQ-BERT using a multi-task MLM objective. Let $\mathcal{C}$ be the set of all $P \times M$ code sequences in our corpus. During training, for each sequence $\mathbf{C}$, we randomly select a set of indices $\mathcal{I}_{\text{mask}}$ to be masked, which constitutes approximately 15% of the $P \times M$ total sub-word positions. The masked input sequence is denoted as $\mathbf{C}_{\text{masked}}$. The model's objective is to predict the original sub-word IDs, $\mathbf{C}_{\text{orig}}$, at these masked positions.

The total loss $\mathcal{L}$ is defined as the sum of the average cross-entropy losses from all $M$ output heads, calculated only over the masked positions. For a single sequence, the loss is:

$$\mathcal{L}(\mathbf{C}) = \sum_{i=1}^{M} \frac{1}{|\mathcal{I}_{\text{mask},i}|} \sum_{(t,i) \in \mathcal{I}_{\text{mask},i}} -\log P(c_{t,i}|\mathbf{C}_{\text{masked}}) \tag{3}$$

where $\mathcal{I}_{\text{mask},i}$ are the masked positions corresponding to the $i$-th subspace, and the probability $P(c_{t,i}|\mathbf{C}_{\text{masked}})$ is computed from the logits produced by the $i$-th output head $O_i$ after a softmax operation. This multi-task setup forces the model to learn the intricate correlations between different subspaces of the reaction vectors. For instance, it might learn that a certain type of reaction in the first 7 dimensions (e.g., related to scalar instructions) often co-occurs with a specific reaction in the last 7 dimensions (e.g., related to memory behavior). This process yields a powerful encoder capable of understanding the deep, compositional grammar of program optimization behavior. The pre-trained Transformer Encoder part of the model is then used to generate embeddings for downstream tasks.

## 3 EXPERIMENTAL SETUP

**Pre-training Dataset.** For self-supervised pre-training, we use a dataset of over 220,000 LLVM IR files constructed for the task of classifying programs by functionalities Mou et al. (2016). Each file

Table 1: Composition of downstream datasets from CompilerGym.

| Uncurated Datasets | | | | | | Curated Datasets | | | |
|---|---|---|---|---|---|---|---|---|---|
| **Type** | **Dataset** | **Train** | **Val** | **Test** | | **Type** | **Dataset** | **Train** | **Val** | **Test** |
| | blas-v0 | 133 | 28 | 29 | | | cbench-v1 | 0 | 0 | 11 |
| | github-v0 | 7,000 | 1,000 | 0 | | | mibench-v1 | 0 | 0 | 40 |
| | linux-v0 | 4,906 | 1,000 | 0 | | Curated | chstone-v0 | 0 | 0 | 12 |
| Uncurated | opencv-v0 | 149 | 32 | 32 | | | npb-v0 | 0 | 0 | 121 |
| | poj104-v1 | 7,000 | 1,000 | 0 | | | | | | |
| | tensorflow-v0 | 415 | 89 | 90 | | | | | | |
| **Total** | – | | | | | | | **19,603** | **3,149** | **335** |

corresponds to a competitive programming solution, providing rich diversity in algorithmic structures and computational patterns. This allows the model to acquire general-purpose program semantics that are independent of specific optimization tasks. We train for 30 epochs with a learning rate of $10^{-4}$ and batch size of 32.

**Downstream Task Datasets.** To evaluate transferability, we adopt benchmarks from Compiler-Gym Cummins et al. (2021b). Importantly, these benchmarks are entirely different from our pre-training corpus, ensuring strict out-of-domain evaluation Fursin (2009); Guthaus et al. (2001); Bailey et al. (1991); Hara et al. (2008); Culjak et al. (2012); Abadi et al. (2016). Following the official split, uncurated benchmarks (e.g., `linux`, `github`) are used for training, while curated benchmarks (e.g., `cbench`, `mibench`) are reserved for testing. Table 1 summarizes the dataset composition.

**Downstream Tasks.** We consider two widely studied tasks in compiler auto-tuning: (1) predicting which optimization pass among 124 candidates yields the largest instruction reduction, and (2) predicting the instruction reduction ratio under the `-Oz` optimization pipeline. These tasks are representative of classification and regression challenges in compiler optimization and capture both fine-grained and holistic optimization effects. Performance is reported using Top-1/Top-5 accuracy and Mean Absolute Error (MAE), respectively.

**Baselines.** We compare against both feature-based and embedding-based methods. Feature-based baselines include **Autophase** Haj-Ali et al. (2020b) (56 handcrafted features) and **InstCount** (LLVM instruction opcode counts), which can be used directly without further training. Embedding-based baselines include **IR2Vec** VenkataKeerthy et al. (2020) and **inst2vec** Ben-Nun et al. (2018), where we use their released pre-trained embeddings. Since inst2vec only provides instruction-level embeddings rather than whole-program representations, we apply an LSTM encoder to aggregate them for downstream tasks. In addition, we include the GNN-based **ProGraML** Cummins et al. (2021a), which is pre-trained on the same algorithm classification dataset as ours, ensuring a fair comparison.

**Implementation Details.** For a fair comparison, all methods are trained on NVIDIA A100 GPUs under the same optimization settings (Adam with learning rate $1 \times 10^{-4}$, batch size 128, 100 epochs). For Best Pass Prediction, the models (except inst2vec) use a two-layer MLP (`512 → 256 → 124`), while the `-Oz` regression task uses a deeper MLP gradually shrinking from 256 to 1 dimension. For `inst2vec`, we directly adopt an LSTM-based model for both downstream tasks, instead of the MLP used by other methods.

## 4 EXPERIMENTS

In this section, we present the empirical evaluation of our proposed behavioral embedding, `Behavioral-PQ`. Our experiments are designed to answer three key research questions:

**RQ1:** Does our proposed behavioral representation outperform state-of-the-art static representations on compiler optimization tasks?

**RQ2:** What is the contribution of each key component in our framework, such as scale-invariant quantification and compositional encoding?

**RQ3:** Does the learned embedding space of our behavioral representation exhibit a meaningful geometric structure that aligns with the semantics of compiler optimization tasks?

Additionally, to evaluate the practical utility and cross-domain generalization of our method, we provide extended analyses on execution runtime, cycle-level performance, and device mapping in Appendix C.

**On reporting validation vs. test performance.** We report both validation and test results. The validation set is not used for model selection or hyperparameter tuning; it reflects the original split and contains programs mostly from the same benchmark families as the training set, leading to generally higher performance. In contrast, the test set is composed mainly of programs from suites outside the training data, providing a stricter out-of-domain evaluation. Thus, validation measures in-distribution generalization, while test primarily assesses cross-domain robustness.

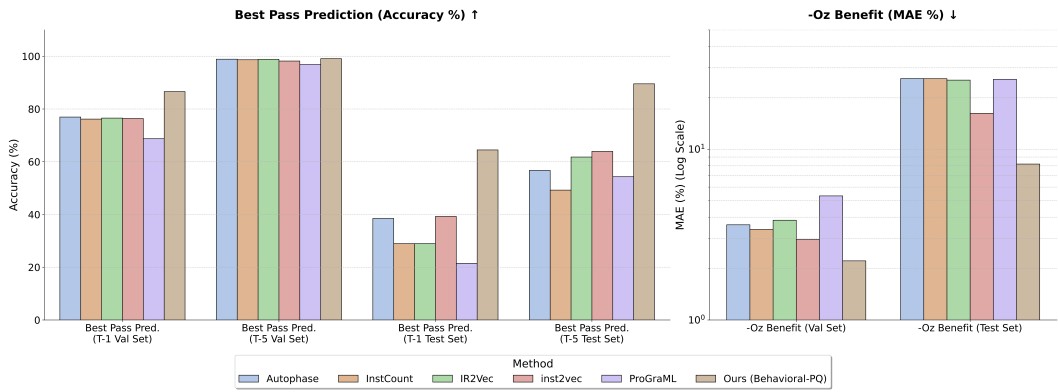

Figure 3: Performance comparison across two prediction tasks: Best Pass Prediction, evaluated in terms of accuracy (higher values indicate better performance), and -Oz Benefit Prediction, evaluated in terms of Mean Absolute Error (lower values indicate better performance).

## 4.1 Main Results: Superior Performance on Downstream Tasks (RQ1)

To evaluate the practical utility and versatility of the learned program representations, we select two downstream tasks that are central to the field of compiler optimization. These tasks were chosen to assess two distinct but equally critical capabilities: fine-grained, single-step decision making (classification) and holistic, long-range performance estimation (regression).

### 4.1.1 Task 1: Best Pass Prediction

This is a 124-class classification problem. We evaluate performance using Top-1 and Top-5 accuracy. **Top-1 accuracy** measures the percentage of cases where the model's single highest-probability prediction is the correct best pass. **Top-5 accuracy** measures the percentage of cases where the correct best pass is included within the model's top five predictions, a metric that reflects the practical utility of the model in narrowing down the search space for autotuners.

Figure 3 presents the results on both the validation and test sets. Our method, `Behavioral-PQ`, achieves a Top-1 accuracy of **64.48 %** and a Top-5 accuracy of **89.55 %** on the held-out test set. This represents a substantial improvement over all baseline methods. Notably, it surpasses the strongest static embedding baseline, inst2vec (39.27% Top-1), by a large margin of over 25 absolute percentage points in Top-1 accuracy. This result strongly suggests that the behavioral spectrum captures critical information about optimization sensitivity that is not readily available in purely static representations.

### 4.1.2 TASK 2: -OZ BENEFIT PREDICTION

We further evaluate the representations on the task of predicting the benefit of the `-Oz` optimization pipeline. This regression task assesses a representation's ability to model the cumulative effect of a long sequence of interacting transformations. The results, measured by Mean Absolute Error (MAE), are presented in Figure 3.

Our `Behavioral-PQ` method achieves a Mean Absolute Error of **8.19%** on the test set, with a corresponding validation MAE of **2.22%**. This is the lowest error among all tested methods. For comparison, the next best baseline, inst2vec, yields an MAE of 16.23%, while other static representations such as IR2Vec and Autophase result in MAEs of 25.40% and 25.92%, respectively. The performance difference between methods is more pronounced on this task than on the single-pass prediction task. The results suggest that while static representations can be effective for modeling immediate, single-pass effects, they are less suited for predicting the outcomes of complex optimization sequences. Our Behavioral Spectrum, by directly encoding the program's reactions to such transformations, appears to provide a more effective signal for this type of long-range predictive task.

Table 2: Ablation study results on both downstream tasks. For Best Pass, Top-5 Acc. (%) is reported. For `-Oz` Pred., MAE (%) is reported. Best test set results for each task are in **bold**.

| Model Variant | Best Pass (Top-5 %) | | -Oz Pred. (MAE %) | |
|---|---|---|---|---|
| | **Validation** | **Test** | **Validation** | **Test** |
| `Ours (KMeans)` | 98.48 | 93.43 | 3.19 | 8.24 |
| `Ours (No-Relative)` | 99.05 | **94.33** | 2.81 | 10.96 |
| `Ours (No-Transformer)` | 98.73 | 87.46 | 2.31 | 10.08 |
| **Ours (Behavioral-PQ, Full)** | **99.08** | 89.55 | **2.22** | **8.19** |

## 4.2 ABLATION STUDIES (RQ2)

To answer **RQ2**, we conduct a series of ablation studies to dissect our framework and validate the contribution of each key design component. We compare our full `Behavioral-PQ` model against three variants: (1) `Ours (KMeans)`, which discards Product Quantization and instead directly clusters the full behavioral vectors using standard K-Means; (2) `Ours (No-Relative)`, which uses absolute feature differences instead of the scale-invariant logarithmic ratio; and (3) `Ours (No-Transformer)`, which removes the Transformer encoder and directly pools the encoded sub-embeddings.

The results are presented in Table 2. For the Best Pass Prediction task, our full `Behavioral-PQ` model achieves the highest Top-5 accuracy on the validation set (99.08%), while the `No-Relative` variant achieves the highest Top-5 accuracy on the test set (94.33%). This suggests that for this classification task, coarser-grained signals captured by absolute differences or monolithic clusters can provide reasonably strong predictive power, although our full model remains competitive on the test set (89.55%).

For the more complex `-Oz` Benefit Prediction task, the superiority of our full model is clear. On both the validation and test sets, `Behavioral-PQ` achieves the lowest MAE (2.22% and 8.19%, respectively), significantly outperforming all ablated versions. Notably, the `No-Relative` and `No-Transformer` variants reach higher MAE values on the test set (10.96% and 10.08%, respectively), confirming that methodscale-invariant quantification is critical for generalization to complex regression tasksmethod, and that the methoddeep contextual reasoning provided by the Transformer is essential for modeling long-range optimization effectsmethod.

## 4.3 EMBEDDING SPACE ANALYSIS (RQ3)

To answer **RQ3**, we investigate whether our learned embedding space exhibits a meaningful geometric structure that aligns with the semantics of compiler optimization. A well-structured space should group programs with similar optimization needs into coherent clusters. We evaluate this property both quantitatively, using a K-Nearest Neighbors (K-NN) classifier, and qualitatively, through t-SNE visualization.

Table 3: Top-1 accuracy (%) of a K-Nearest Neighbors classifier (k=5) on the Best Pass Prediction task, evaluated on the test set. This metric reflects the semantic structure of each embedding space.

| Embedding Method | Autophase | InstCount | IR2Vec | inst2vec | ProGraML | Ours |
|---|---|---|---|---|---|---|
| K-NN Top-1 Acc. (%) | 46.57 | 75.82 | 74.33 | 72.15 | 70.75 | **79.70** |

The results provide converging evidence of our method's superiority. **Quantitatively**, we use a K-NN classifier ($k = 5$), whose performance directly reflects the local semantic coherence of a space. As shown in Table 3, our `Behavioral-PQ` embedding achieves a Top-1 accuracy of **79.70%**, significantly outperforming all baselines, including the next best method, InstCount (75.82%). **Qualitatively**, this strong result is visually corroborated by the t-SNE projection of the embedding spaces, presented in Figure 4. The visualization shows that our `Behavioral-PQ` space is the only one to exhibit distinct, well-separated clusters of programs that share the same optimal pass. For instance, programs for which `-instcombine` is optimal are naturally grouped together. Together, these quantitative and qualitative results suggest that our behavioral approach generally learns a representation space that is reasonably well-structured and semantically aligned with the task of compiler optimization.

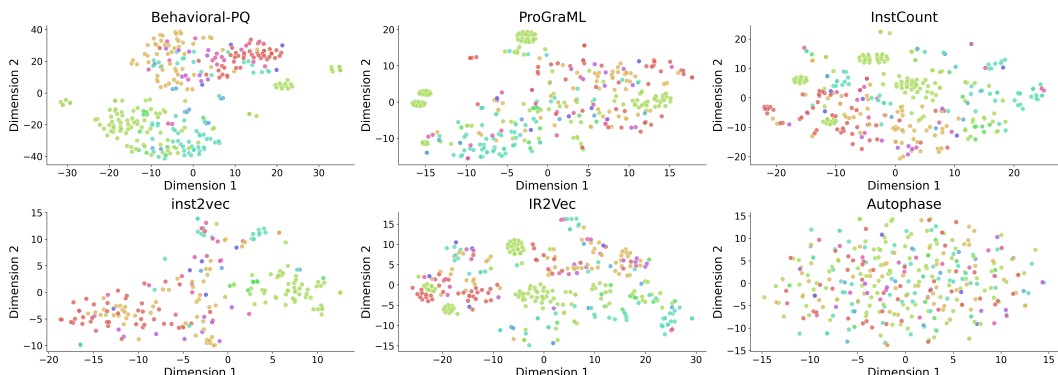

Figure 4: t-SNE visualization of embedding spaces for the test set. Each point is a program, colored by its true best optimization pass.

# 5 RELATED WORK

Program representations for machine learning in compiler optimization have evolved from handcrafted features to learned embeddings Zhu et al. (2024); Cummins et al. (2023); Gong et al. (2025); Liu et al. (2021); Park et al. (2022). Early approaches relied on manually designed IR-level metrics such as Autophase Haj-Ali et al. (2020b) and InstCount Lattner & Adve (2004); Cummins et al. (2021b), which count instructions, branches, and memory operations. These features are simple and interpretable but lack the expressiveness needed to capture complex behaviors, limiting generalization across diverse code bases.

Learned representations instead encode semantics automatically. `inst2vec` Ben-Nun et al. (2018) applies skip-gram models on LLVM IR, capturing local context but ignoring control/data flow. IR2Vec VenkataKeerthy et al. (2020) extends this via graph-based flow analysis, while `ProGraML` Cummins et al. (2021a) unifies control, data, and call graphs in a multigraph for message passing, enabling richer analyses. These approaches better capture non-local dependencies and are less sensitive to superficial variations. Dynamic profiling has also proven useful Xu et al. (2023); Duesterwald et al. (2003), recording runtime behaviors such as memory accesses and input/output values. Such signals complement static embeddings, motivating quasi-dynamic approaches that integrate both perspectives for more effective optimization.

## 5.1 LIMITATIONS AND FUTURE WORK

**Limitations.** Our approach has four main limitations. First, the diversity of optimization probes may be insufficient for some program classes, although the selected probes maintain reasonably good optimization performance. Second, inference requires computing $P + 1$ Autophase features per program, introducing some preprocessing overhead (about 0.2s per program), although this remains faster and more stable compared with collecting full dynamic features. Third, our evaluation is currently focused on compiler optimization tasks,

with limited validation on other downstream tasks such as program classification. Fourth, while behavioral vocabularies provide interpretability, their semantic meaning and coverage are still limited, which may affect the generalizability of the representations.

**Future Work.** Future directions include developing adaptive probe selection strategies to better suit different program classes, reducing preprocessing costs through more efficient Autophase computation, and exploring limited integration of dynamic information to improve prediction accuracy. Additionally, we plan to validate the approach on a broader range of downstream tasks beyond compiler optimizations, and to enhance the interpretability and coverage of behavioral vocabularies for more explainable program representations.

## 6 CONCLUSION

This work introduces a quasi-dynamic paradigm for program representation, termed as **Behavioral Embeddings**, which characterize programs by modeling their responses to a series of carefully designed optimization probes. By capturing these optimization-sensitive behaviors, the approach encodes deep semantics that are difficult to extract from static structure alone. Compared with purely static embeddings, it effectively balances the efficiency of static analysis with the richer, task-relevant insights typically provided by dynamic profiling. Empirical results demonstrate clear advantages: our model substantially improves accuracy on best pass prediction and reduces error in optimization benefit prediction.

## ETHICS STATEMENT

The research presented in this paper adheres to ethical principles for academic work. All program corpora used for pre-training and evaluation, such as the OJ dataset and CompilerGym benchmarks, are derived from publicly available, open-source codebases intended for research purposes. We acknowledge that the large-scale pre-training of our Transformer models requires significant computational resources, which has an associated environmental impact. We argue that this cost is justified by the foundational nature of this research, which aims to establish a new, more efficient paradigm for compiler optimization that could, in the long term, reduce overall computational waste. While our work focuses on benign compiler optimization, we recognize that advanced program representation techniques could potentially be misused for analyzing or optimizing malicious software. However, our proposed method does not introduce any capabilities uniquely suited for such applications beyond those already present in existing program analysis tools. We advocate for the responsible use of these technologies within the academic and industrial communities.

## REPRODUCIBILITY

To support the reproducibility of Behavioral-PQ, we make the complete source code and experimental configuration publicly accessible. All models, training datasets, and scripts can be found at: https://anonymous.4open.science/r/PREP-311F/. The repository provides step-by-step instructions for setting up the environment, running the experiments, and reproducing the results on standard benchmarks. By providing these resources, we aim to enable independent verification and replication of our findings, fostering further progress in compiler optimization research.

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

APPENDIX

## A  DETAILED CASE STUDY ON `BLAS-V0_127.LL`

To provide a more concrete, micro-level example of our model's behavior, we conduct a deep-dive analysis into a successful prediction case: the program `blas-v0_127.ll` from the test set.

**Scenario Overview.**  For this program, the empirically determined single best optimization pass is `-instcombine`. Our `Behavioral-PQ` based model successfully includes this pass in its Top-5 predictions, with `-instcombine` being the highest-ranked (Top-1) recommendation. It is worth noting that a baseline model relying solely on static Autophase features failed to include `-instcombine` in its Top-5 predictions for this case. The respective predictions are shown in Table 4.

Table 4: Model predictions for the case study program `blas-v0_127.ll`.

| Item | Details |
|------|---------|
| Ground Truth Best Pass | `-instcombine` (ID: 53) |
| **Our Model's Top-5** | **`-instcombine`**, `-early-cse-memssa`, `-ipsccp`, etc. |

**Analyzing the Relationship between Predictions and Probe Reactions.**  To better understand the information available to our model, we investigate the relationship between the program's Behavioral Spectrum and the model's correct prediction. We first identify which of the 100 probes elicits the strongest reaction. The *reaction strength* is defined as the magnitude of reduction in a key Autophase feature (feature #51).

Our analysis shows that **Probe #10**, a complex 50-pass sequence, elicited the single strongest reaction. We then cross-reference our model's Top-5 predicted passes against the composition of this most impactful probe sequence.

The results of this analysis are detailed in Table 5. We found that the 50 passes within the strongest probe sequence contain a total of **12 instances** of passes that were also present in our model's Top-5 prediction list. Most notably, the ground truth best pass, `-instcombine`, appears **3 times** within this single most reactive probe.

Table 5: Analysis of the alignment between the model's Top-5 predictions and the composition of the most reactive probe sequence (Probe #10) for `blas-v0_127.ll`.

| Analysis Item | Finding |
|---------------|---------|
| Most Reactive Probe Index | Probe #10 |
| Model's Top-5 Predictions | `-instcombine` (Top-1), `-early-cse-memssa`, `-ipsccp`, `-globalopt`, `-mergefunc` |
| **Occurrences of Top-5 Passes within Probe #10** | |
| `-instcombine` | **3 times** |
| `-early-cse-memssa` | 4 times |
| `-ipsccp` | 2 times |
| `-globalopt` | 3 times |
| `-mergefunc` | 0 times |
| **Total Alignment** | **12 out of 50 passes** in the strongest probe are from the model's Top-5 list. |

**Discussion.**  This analysis reveals a strong correlation: the program reacts most intensely to a probe sequence that is richly populated with passes the model identified as highly effective. While this correlation does not establish a direct causal link—as the probe's overall effect is a result of the entire 50-pass sequence, not just the individual passes—it does provide valuable insight. It suggests that the Behavioral Spectrum contains discernible signals related to the efficacy of certain optimization types. The pre-trained `PQ-BERT` model appears to be capable of identifying these signals within the complex, high-dimensional spectrum and associating them with correct individual pass recommendations.

Table 6: Performance comparison with general-purpose Large Language Models (LLMs) in a zero-shot setting. For the Best Pass Prediction task, we report Top-1 and Top-5 accuracy (%). For the `-Oz` Benefit Prediction task, we report Mean Absolute Error (MAE, %). The performance of our specialized model and the best traditional baseline are included for reference.

| Model / Method | Task 1(Top-1 %) | Task 1(Top-5 %) | Task 2 |
|---|---|---|---|
| *General-Purpose Large Language Models (Zero-Shot)* | | | |
| gpt-5-mini | 35.52 | 56.12 | 23.87 |
| zai-org/GLM-4.5 | 34.93 | 59.10 | 22.81 |
| DeepSeek-V3 | 34.03 | 54.93 | 24.05 |
| baidu/ERNIE-4.5-300B-A47B | 2.39 | 23.28 | 23.70 |
| tencent/Hunyuan-A13B-Instruct | 1.30 | 2.60 | 23.52 |
| Tongyi-Zhiwen/QwenLong-L1-32B | 23.54 | 48.25 | 22.74 |
| *Specialized Models (for reference)* | | | |
| Autophase (Best Baseline) | 38.51 | 56.72 | 25.92 |
| **Ours (Behavioral-PQ)** | **64.48** | **89.55** | **8.19** |

## B  COMPARISON WITH GENERAL-PURPOSE LARGE LANGUAGE MODELS

Recent advancements in Large Language Models (LLMs) have demonstrated remarkable capabilities in code understanding and generation. To situate our work within this modern context, we conducted an experiment to evaluate the zero-shot performance of several state-of-the-art general-purpose LLMs on our two downstream tasks. We provided the models with the full LLVM IR, its Autophase features, and the list of candidate passes, then prompted them to return their predictions in a structured JSON format.

The results, summarized in Table 6, reveal a clear trend. While some of the best-performing LLMs (e.g., `gpt-5-mini`, `GLM-4.5`, `DeepSeek-V3`) achieve a respectable Top-5 accuracy of around 55-60% on the Best Pass Prediction task, their performance is still substantially lower than our specialized `Behavioral-PQ` model (89.55%). Notably, even the best LLM's Top-1 accuracy (35.52%) does not surpass that of the simple, handcrafted Autophase baseline (38.51%). On the more complex `-Oz` Benefit Prediction task, the performance gap is even more stark. The best-performing LLM (`GLM-4.5`) yields an MAE of 22.81%, an error rate nearly three times higher than that of our method (8.19%).

These findings provide a crucial insight: while general-purpose LLMs possess a broad understanding of code, this knowledge does not readily translate to the highly specialized, quantitative, and nuanced domain of compiler optimization. Their zero-shot reasoning struggles to match the performance of a smaller, domain-specific model that has been pre-trained on a representation—our Behavioral Spectrum—that is intrinsically aligned with the task of predicting optimization outcomes.

## C  EXTENDED GENERALIZATION AND EMPIRICAL ANALYSIS

To further validate the robustness, practical utility, and across-task generality of our proposed Behavioral-PQ embeddings, we present additional experimental results. These evaluations extend beyond the code-size metrics used in the main text to include execution runtime, cycle-level performance, and cross-domain generalization tasks.

### C.1  EVALUATION ON RUNTIME-RELATED METRICS

While instruction-count reduction serves as a stable proxy for optimization quality, execution runtime is the ultimate performance metric in production environments. We conducted two new experiments to evaluate how well our embeddings capture runtime-related characteristics.

**Task A: Predicting -O3 Runtime on NeuroVectorizer Haj-Ali et al. (2020a).** We formulated a regression task to predict the absolute execution time of programs after -O3 optimization. As shown in Table 7, Behavioral-PQ achieves the lowest MAE and the highest $R^2$ among all tested static representations.

**Task B: Predicting Cycle-Reduction Percentage via LLVM-MCA.** We evaluated the percentage of cycle reduction as estimated by `llvm-mca`, a static analysis tool that models the execution of machine code on a specific CPU microarchitecture. This metric provides a high-fidelity simulation of hardware-level execution efficiency and pipeline utilization without requiring physical hardware execution. As shown in Table 8, our method consistently outperforms static baselines in predicting these simulated microarchitectural performance gains under the -O3 optimization pipeline.

Table 7: Performance on predicting absolute execution time (NeuroVectorizer dataset).

| Representation | MAE $\downarrow$ | $R^2 \uparrow$ |
|---|---|---|
| IR2Vec | 94.11 | 0.746 |
| Autophase | 86.87 | 0.808 |
| ProGraML | 89.50 | 0.741 |
| **Ours (Behavioral-PQ)** | **85.54** | **0.807** |

Table 8: Performance on predicting cycle reduction percentage (CompilerGym).

| Representation | MAE (%) $\downarrow$ |
|---|---|
| IR2Vec | 10.86% |
| ProGraML | 10.34% |
| Autophase | 8.85% |
| **Ours (Behavioral-PQ)** | **8.78%** |

## C.2 COMPARISON WITH DYNAMIC BASELINES

A key advantage of our quasi-dynamic approach is its ability to capture behavioral signals without the overhead of physical execution. To position our method, we compared it against a purely dynamic baseline using hardware performance counters (HPCs) collected via `linux perf`.

For this baseline, we extracted a 28-dimensional dynamic feature vector during program execution. This vector consists of 20 raw hardware event counts and 8 derived performance ratios, providing a comprehensive profile of the program's runtime behavior.

**Dynamic Feature Composition (28 Dimensions):**

- **Raw Hardware Events (20):** We tracked fundamental metrics including total cycles, instructions retired, branches, branch-misses, frontend/backend stalled cycles, cache references/misses (L1-dcache, L1-icache, LLC), and Translation Lookaside Buffer (TLB) loads/misses (dTLB, iTLB), as well as system-level events like context-switches and CPU-migrations.

- **Derived Performance Ratios (8):** To capture architectural efficiency beyond absolute counts, we calculated eight ratio-based features: Instructions Per Cycle (IPC), branch miss rate, L1-dcache miss rate, L1-icache miss rate, LLC miss rate, dTLB miss rate, iTLB miss rate, and the frontend stall ratio.

As shown in Table 9, despite having access to these high-fidelity execution profiles, the dynamic baseline achieves an MAE of 116.83 ($R^2 = 0.726$), which is higher than our Behavioral-PQ's MAE of 85.54 ($R^2 = 0.807$). This result underscores that optimization sensitivity (how a program *reacts* to changes) is a more robust indicator of post-optimization performance than a single-point runtime snapshot, as raw HPCs can be susceptible to environmental noise and exhibit limited predictive power for complex code transformations.

Table 9: Comparison with 28-dimensional dynamic HPC baseline on -O3 runtime prediction.

| Representation | Type | MAE $\downarrow$ | $R^2 \uparrow$ |
|---|---|---|---|
| Dynamic (28-dim HPC) | Dynamic | 116.83 | 0.726 |
| **Ours (Behavioral-PQ)** | **Quasi-dynamic** | **85.54** | **0.807** |

## C.3 CROSS-TASK GENERALITY: HETEROGENEOUS DEVICE MAPPING

To evaluate whether our representation captures general program semantics beyond compiler optimization, we applied it to the Heterogeneous Device Mapping (DevMap) Cummins et al. (2017) task. This task requires predicting whether an OpenCL kernel executes faster on a CPU or a GPU, which demands a deep understanding of code-hardware interactions.

As shown in Table 10, our method achieves the best performance (68.38% accuracy) among all baseline embeddings. This demonstrates that optimization-sensitivity probes can elicit fundamental semantic features that generalize to other systems-level tasks.

## C.4 COMPARISON WITH FINE-TUNED LLMs AND IR-BASED METHODS

We further compared our approach against modern Large Language Models (LLMs) and specialized IR-based models under the *pre-train + fine-tune* paradigm. We fine-tuned Qwen2.5 (at various scales) and IRCoder Paul et al. (2024) on the DevMap task.

Table 10: Comparison with modern models on the DevMap classification task.

| Model / Representation | Type | Accuracy ↑ |
|---|---|---|
| IR2Vec | Static Embedding | 64.71% |
| Autophase | Static Features | 66.91% |
| ProGraML | GNN-based | 62.26% |
| Qwen2.5-1.5B (Finetuned) | LLM | 58.82% |
| Qwen2.5-3B (Finetuned) | LLM | 66.18% |
| IRCoder (starcoderbase-1B) | IR-Pretrained | 61.03% |
| IRCoder (starcoderbase-7B) | IR-Pretrained | 60.29% |
| **Ours (Behavioral-PQ)** | **Quasi-dynamic** | **68.38%** |

The results in Table 10 highlight that while models like IRCoder excel at code generation by treating IR as static text, they may struggle to capture the nuanced behavioral changes critical for performance-sensitive tasks. In contrast, Behavioral-PQ's focus on optimization sensitivity provides a more effective representation for these domains, even when compared to significantly larger models. This confirms that our quasi-dynamic paradigm is complementary to existing IR-based static methods.

## D THE USE OF LARGE LANGUAGE MODELS

Large Language Models were utilized to support the enhancement of clarity and coherence throughout the manuscript. They helped with rephrasing, maintaining academic standards, and improving overall readability. It is important to note that their role was confined to the writing process, and all material was carefully reviewed and finalized by the authors themselves.

