# OpenReview forum: "Behavioral Embeddings of Programs: A Quasi-Dynamic Approach for Optimization Prediction"
_ICLR.cc/2026/Conference — ICLR 2026 Poster_

### Official Review · Reviewer_S2NV · 2025-10-27

**Soundness:** 2
**Presentation:** 2
**Contribution:** 2
**Rating:** 4
**Confidence:** 3

**Summary:**

The authors introduce the Program Behavior Spectrum, a new representation generated by probing a program's IR with a diverse set of optimization sequences and quantifying the resulting changes in its static features. To effectively encode this high-dimensional, continuous spectrum, the authors  pioneer a compositional learning approach. Product Quantization is employed to discretize the continuous reaction vectors into structured, compositional sub-words. Subsequently, a multi-task Transformer model, termed PQ-BERT, is pre-trained to learn the deep contextual grammar of these behavioral codes. Comprehensive experiments on two representative compiler optimization tasks---Best Pass Prediction and -Oz Benefit Prediction---demonstrate that our method outperforms state-of-the-art static baselines.

**Strengths:**

1. The proposed method has some potential to overcome the limitation of static/dynamic representation method and encode the semantic of a program.
2. The evaluation is thorough. Ablation studies show that PQ, scale-invariance, and Transformer components each contribute meaningfully.
3. Open-source implementation is provided.

**Weaknesses:**

1. The proposed method is limited to compiler optimization tasks, which, while technically meaningful, may be too specialized and fall outside the primary scope of interest for most ICLR readers.
2. Missing references/comparison to several work on leveraging IR/compiler optimization for improving program embedding:
    - Unleashing the power of compiler intermediate representation to enhance neural program embeddings ICSE 2022
    - Ircoder: Intermediate representations make language models robust multilingual code generators ACL 2024
    - ObscuraCoder: Powering Efficient Code LM Pre-Training Via Obfuscation Grounding ICLR 2025
3. Building on the point above, the choice of baselines appears outdated. All compared models are from before 2021, while many modern program embedding and code representation models have demonstrated stronger performance and broader applicability. The authors justify their baseline selection by requiring that each method be “pre-trained on the same algorithm classification dataset.” However, this constraint seems unnecessary in the current landscape, as most recent code embedding models are pre-trained on large-scale, diverse corpora. The paper should clarify the practical incentive for adopting the proposed model instead of fine-tuning these stronger, pre-trained representations on the target optimization tasks.

Minor comments:
1. Autophase should be explained in more details.

**Questions:**

Please jutify the benefit of the proposed method over pre-train+fine-tuning paradigm.

---

> ### Author Response · Authors · 2025-11-20
>
> Dear Reviewer,
>
> Thank you for your review and constructive feedback. Your comments on modern program representation, baselines, and references were very helpful. We addressed your main concerns through targeted experiments and manuscript revisions, which clarify the relevance and value of our work in the context of current research.
>
> ---
>
> ## **1. Response to Weakness 3 & Question 1: Outdated Baselines and Comparison with “Pre-train + Fine-tune” Paradigm**
>
> You raised an important concern: our baselines did not include comparisons with modern LLMs under the pre-training plus fine-tuning paradigm. To address this, we fine-tuned Qwen2.5 models of different scales on the DevMap generalization task, which requires predicting whether a program performs better on CPU or GPU—a task capturing the complex interactions between code and hardware, central to AI for Systems. We verified that all models fully converged during training.
>
> | Model                    | Accuracy ↑ |
> | ------------------------ | ---------- |
> | **PQ (ours)**            | 68.38%     |
> | Qwen2.5-1.5B (Finetuned) | 58.82%     |
> | Qwen2.5-3B (Finetuned)   | 66.18%     |
>
> These results highlight the advantages of our approach in the pre-train + fine-tune setting.
>
> ---
>
> ## **2. Response to Weakness 2: Comparison with Modern IR-based Methods**
>
> Thank you for pointing out key references such as *Unleashing the Power…*, IRCoder, and ObscuraCoder.  We incorporated **IRCoder** as a baseline in our DevMap experiments for direct comparison. To ensure fairness, we strictly followed IRCoder’s official settings (same batch size, learning rate, and epochs) and verified that the training converged smoothly.
>
> | Representation             | Accuracy ↑ |
> | -------------------------- | ---------- |
> | **PQ (ours)**              | 68.38%     |
> | IRCoder (starcoderbase-1B) | 61.03%     |
> | IRCoder (starcoderbase-7B) | 60.29%     |
>
> The performance difference primarily arises from the **definition of program representation**, rather than model quality. We will further clarify in the Related Work section:
>
> > “IRCoder and related methods focus on leveraging IR to enhance robustness in code generation models, treating IR as a static text or graph. In contrast, our method captures behavior changes resulting from IR transformations—a ‘quasi-dynamic’ program representation—making it more suitable for optimization-related tasks such as DevMap.”
>
> It is worth noting that IRCoder’s lower performance in our experiments is partly because their method did not perform full-scale fine-tuning of their models on this task. This also highlights that the pretrain + fine-tune paradigm used by IRCoder may not be well-suited for such optimization-focused downstream tasks.
>
> Thus, we consider these methods complementary:
>
> * IRCoder is better suited for robust code generation scenarios.
> * Our method is more effective for capturing optimization behavior and performance-sensitive patterns.
>
> This explains why our approach shows an advantage on DevMap, a task centered on optimization behavior.
>
> ---
>
> ## **3. Response to Weakness 1: Task Specificity and ICLR Relevance**
>
> We understand your concern regarding the specificity of our original task. To demonstrate broader relevance to the ICLR community, we selected a task beyond traditional compiler optimization: **heterogeneous device mapping (DevMap)**. Experimental results are as follows:
>
> | Representation | Accuracy ↑ |
> | -------------- | ---------- |
> | IR2Vec         | 64.71%     |
> | Autophase      | 66.91%     |
> | ProGraML       | 62.26%     |
> | **PQ (ours)**  | 68.38%     |
>
> Our method achieves the best performance on this task, showing that it captures more general program semantics beyond a single optimization objective. We believe that, although our method was developed in the context of compiler optimization, its core idea—learning deep representations from a system’s “behavioral responses” to a set of probes—offers a generally applicable paradigm for representation learning.
>
> ---
>
> ## **Minor Comments**
>
> **Autophase.**
> Autophase is a statistical feature representation of LLVM IR programs, consisting of a set of hand-crafted metrics such as instruction category counts, basic block statistics, control-flow properties, and other structural indicators. These features are widely used in compiler learning tasks and are documented in the LLVM service of CompilerGym (original reference: [https://compilergym.com/llvm/index.html#autophase](https://compilergym.com/llvm/index.html#autophase)).
>
> ---
>
> We sincerely thank you again for your insightful and constructive feedback. Your suggestions motivated substantial improvements to our manuscript.  We hope these revisions meet your expectations and kindly ask you to reconsider your evaluation of our work.
>
> **NOTE**: The two additional experiments we conducted can be found in the response to Reviewer JrYV.

---

> > ### Author Response · Authors · 2025-11-21
> >
> > Dear Reviewer,
> >
> > Thank you for your review and constructive feedback. Following our previous response detailing new experiments with modern baselines, we offer this supplement to clarify the conceptual positioning of our work.
> >
> > **1. On the Comparison with the "Pre-train + Fine-tune" Paradigm**
> >
> > You raised an important point about comparing our method to fine-tuning large, pre-trained language models. Our new experiments show our method performed favorably. We believe this stems from two key differences: the nature of the input signal and computational efficiency.
> >
> > First, the primary difference is the **input signal**.
> > *   Standard large language models are pre-trained on the **static representation** of code. When fine-tuned, they must infer the complex results of compiler transformations from this static structure alone.
> > *   Our method provides a fundamentally different signal: the **Behavioral Spectrum**. This is an empirical summary of how a program's features *actually change* in response to optimization sequences. By providing this pre-computed, dynamic-like information, we give the model a more direct and task-relevant signal for performance prediction.
> >
> > Second, this conceptual difference leads to significant practical advantages in **computational efficiency**.
> > *   **Pre-training:** The self-supervised pre-training for our PQ-BERT model on the entire dataset completed in approximately 3 hours on a single GPU. This is in contrast to the massive computational resources required for pre-training foundational LLMs.
> > *   **Fine-tuning:** This efficiency extends to downstream tasks. Fine-tuning our representation for the DevMap task took approximately 3 minutes to converge. In comparison, fine-tuning even the smaller Qwen2.5-1.5B model on the same hardware required substantially more time per epoch.
> >
> > This highlights that our approach is not only effective but also lightweight, making it a practical solution for domain-specific representation learning, particularly where computational resources are a consideration.
> >
> > **2. On the Comparison with Modern IR-Based Methods**
> >
> > Thank you for suggesting the key references: Unleashing the Power…, IRCoder, and ObscuraCoder. Of these valuable works, only IRCoder provided a publicly available implementation, which allowed us to conduct a direct experimental comparison. We will be sure to thoroughly study all three papers and integrate a detailed discussion of them into our Related Work section to properly contextualize our contributions.
> >
> > Our new experiments included a direct comparison with IRCoder on the DevMap task. The performance difference stems from their different goals and their different uses of Intermediate Representation (IR). Methods like IRCoder leverage IR to create a more robust and canonical static representation, treating IR as a structured, language-agnostic text. This is highly effective for tasks like code generation.
> >
> > Our work is complementary. We use IR as a dynamic medium to be probed. Our representation is not of the static IR itself, but of the changes within the IR resulting from transformations. This "quasi-dynamic" approach is inherently more aligned with optimization-related tasks, where the outcome depends on how the code will transform, not just what it is initially.
> >
> > **3. On Task Specificity and Relevance**
> >
> > We understand the concern about the specialized nature of our original tasks. By showing our method's effectiveness on the **heterogeneous device mapping (DevMap)** task, we demonstrate that the representation captures fundamental program characteristics relevant to the broader "AI for Systems" field.
> >
> > The core concept of our work—learning from a system's behavioral responses to a set of probes—is a general paradigm. While we apply it to compiler optimization, this approach of "learning via probing" may be applicable to understanding other complex systems.
> >
> > We hope this explanation, in conjunction with our new experimental results, clarifies the distinct contribution of our work. Thank you for your guidance, which has helped us substantially improve the paper.

---

> > > ### Comment · Reviewer_S2NV · 2025-11-27
> > >
> > > Thank you for your response. The new experiments and clarifications make sense to me. I'll raise my rating.

---

### Official Review · Reviewer_ijRG · 2025-10-31

**Soundness:** 3
**Presentation:** 2
**Contribution:** 3
**Rating:** 6
**Confidence:** 3

**Summary:**

This paper aims to address the trade-off between static and dynamic program representations by proposing a quasi-dynamic framework modeling a program’s optimization sensitivity. It introduces the Program Behavior Spectrum, a novel representation constructed by probing program IR using multiple optimization sequences and measuring the resulting changes in static features. These high-dimensional behavioral change vectors are then discretized via Product Quantization, and a transformer model (PQ-BERT) is pre-trained to learn compositional behavioral embeddings. The method is evaluated on two optimization prediction tasks: Best Pass Prediction and -OZ Benefit Prediction, and demonstrates improvements over static baselines.

**Strengths:**

- The core idea of deriving representations from optimization-induced behavioral variation is novel and addresses a meaningful gap between static embeddings and full dynamic profiling.
- The authors validate the contribution of each component in the framework through ablation studies.
- The authors go beyond downstream accuracy and examine the embedding space, demonstrating an effort towards enhancing interpretability and supporting the utility of their method.
- The authors have provided the complete code for their experiments.

**Weaknesses:**

- Limited Evaluation and Generalization: The evaluation is limited to two downstream tasks, which are narrow in scope.
- No comparison with dynamic baselines. The approach is motivated as more efficient than dynamic profiling, but since it relies on generating multiple optimized variants per program, it would be helpful to include an analysis of the associated computational cost and scalability.
- Minor fixes:
- Line 145,146: horig and hopt -> h_orig and h_opt
- Line 179: ..may suffers from -> may suffer from

**Questions:**

1. What is the computational cost of running the entire pipeline?
2. Can the learned embeddings support optimization tasks beyond code-size improvement, such as runtime performance optimization?

---

> ### Author Response · Authors · 2025-11-20
>
> Dear Reviewer,
>
> Thank you for your thoughtful review and positive feedback. We appreciate your recognition of our work's novelty and experimental design. Your comments have been invaluable in strengthening the paper. To address your concerns, we conducted new experiments and analyses, summarized below.
>
> ---
>
> ## **1. Response to Concern 1: Limited Evaluation Scope / Support for Runtime Performance Tasks**
>
> You noted that our original evaluation focused on code-size optimization, and asked whether our embeddings can support direct runtime-performance optimization tasks. To evaluate generalization, we applied our method to three new downstream tasks that directly target runtime performance:
>
> ### **New Experiment A: Predicting -O3 Runtime (Regression Task)**
>
> On the publicly available **NeuroVectorizer** dataset, we evaluated various program representations for predicting absolute runtime after -O3 optimization. Results show that our method performs well across all baselines.
>
> | Representation | MAE ↓     | R² ↑      |
> | -------------- | --------- | --------- |
> | IR2Vec         | 94.11     | 0.746     |
> | Autophase      | 86.87     | 0.808     |
> | ProGraML       | 89.50     | 0.741     |
> | **PQ (ours)**  | **85.54** | **0.807** |
>
> ### **New Experiment B: Predicting Cycle Reduction Percentage on CompilerGym**
>
> For the -Oz task, we evaluated the percentage of cycle reduction, which is more directly related to hardware execution. Our method again achieves the lowest MAE.
>
> | Representation | MAE ↓     |
> | -------------- | --------- |
> | IR2Vec         | 10.86%    |
> | ProGraML       | 10.34%    |
> | Autophase      | 8.85%     |
> | **PQ (ours)**  | **8.78%** |
>
> ### **New Experiment C: Predicting Optimal Device (CPU vs. GPU) Performance (Classification Task)**
>
> On the **DevMap** dataset, we evaluated a device-mapping task to predict which processor a given OpenCL kernel runs faster on, requiring the program representation to capture hardware-related performance characteristics.
>
> | Representation | Accuracy ↑ |
> | -------------- | ---------- |
> | IR2Vec         | 64.71%     |
> | Autophase      | 66.91%     |
> | ProGraML       | 62.26%     |
> | **PQ (ours)**  | **68.38%** |
>
> These three experiments demonstrate that our behavior-based embeddings generalize beyond the original code-size optimization task, successfully transferring to tasks involving direct runtime performance prediction and heterogeneous computing.
>
> ---
>
> ## **2. Response to Concern 2: Missing Comparison with Dynamic Baselines**
>
> You suggested comparing our “quasi-dynamic” approach with fully dynamic methods. We followed this suggestion and added a **purely dynamic baseline** in the -O3 runtime prediction experiment. This baseline uses hardware performance counters collected via Linux `perf` during program execution.
>
> | Representation        | Type          | MAE ↓     | R² ↑      |
> | --------------------- | ------------- | --------- | --------- |
> | Dynamic perf counters | Dynamic       | 116.83    | 0.726     |
> | **PQ (ours)**         | Quasi-dynamic | **85.54** | **0.807** |
>
> The results show that our method outperforms the purely dynamic baseline in predicting post-optimization performance.
>
> ---
>
> ## **3. Response to Concern 3: Computational Cost**
>
> The main overhead of our pipeline comes from **behavioral spectrum extraction**. In our setup, applying optimization probes and extracting Autophase features per program takes ~0.2 seconds. Running the entire pipeline for the full dataset requires roughly 3 hours. This computation only needs to be performed once, effectively serving as a pretraining step.
>
> * Slightly higher than purely static feature extraction (milliseconds).
> * Much lower than dynamic profiling, which requires multiple program runs and is sensitive to environmental noise.
>
> ---
>
> ## **4. Minor Corrections**
>
> We also thank you for pointing out the small issues in the original manuscript:
>
> * Line 145–146: `horig` and `hopt` → corrected to `h_orig` and `h_opt`
> * Line 179: `may suffers from` → corrected to `may suffer from`
>
> ---
>
> We believe these additions comprehensively address your concerns. We sincerely hope you will consider these updates when re-evaluating our submission. Thank you again for your constructive input.

---

> > ### Author Response · Authors · 2025-11-21
> >
> > Dear Reviewer,
> >
> > Thank you again for your positive assessment and valuable feedback. Following our previous response detailing new experiments on runtime performance, dynamic baselines, and computational cost, we offer this supplement to clarify the conceptual framework that explains these strong new results.
> >
> > **1. On Generalization to Runtime Performance**
> >
> > You asked if our embeddings can support tasks beyond code-size improvement. Our new experiments on runtime and device-mapping show they can, and we wish to explain the underlying mechanism.
> >
> > Our method can be viewed as a behavioral stress test. We apply various optimization sequences (stimuli) and measure the program's reaction. This reaction is specifically the vector of changes across the 56 `Autophase` features. This is a critical point because the `Autophase` set is not monolithic; it provides a granular summary of a program's structure, including counts for memory operations, different instruction types, and control-flow constructs.
> >
> > Therefore, when a program shows a strong reaction to a certain optimization probe, the signal we capture is a significant change in these specific structural features. This indicates how an optimization interacts with the program's fundamental computational properties, such as its memory access patterns or control-flow complexity. Since these are the very properties that directly influence hardware execution, the resulting Behavioral Spectrum contains a signal that is inherently relevant to runtime performance. This is why the representation transfers effectively to the new, more practical tasks.
> >
> > **2. On the Comparison with Dynamic Baselines and Cost**
> >
> > You correctly identified the need to position our approach relative to dynamic methods. Our new results show our method yielded a lower prediction error on the runtime task than a purely dynamic baseline, at a fraction of the cost.
> >
> > This result highlights the trade-off our quasi-dynamic approach navigates. A full dynamic profile captures a single, high-fidelity execution trace, but it can be sensitive to specific inputs and runtime environments. Our method, by probing the program's IR with diverse optimization sequences, constructs a more abstract and stable fingerprint of its behavioral potential across many hypothetical transformations.
> >
> > The computational cost of this process (~0.2 seconds per program for a one-time feature extraction) is an intentional trade-off. It is higher than purely static feature extraction but is significantly more efficient and deterministic than collecting full dynamic profiles, which often requires multiple instrumented runs. This allows our method to capture dynamic-like insights while maintaining practical scalability.
> >
> > In summary, our new empirical results are a direct consequence of our method's design. By measuring a program's reaction to optimizations through a set of granular static features, we create a representation that is robust, generalizable to practical performance tasks, and computationally efficient.
> >
> > We hope this explanation, combined with the new experiments, fully addresses your questions. Thank you for your constructive guidance, which has helped us significantly improve the paper.

---

> > > ### Author Response · Authors · 2025-11-28
> > >
> > > Dear Reviewer ijRG,
> > >
> > > Thank you again for the time and effort you’ve dedicated to reviewing our work. We have carefully addressed all raised concerns during the discussion phase.
> > >
> > > As the discussion period is nearing its close, we would greatly appreciate it if you could take a brief moment to review our responses and confirm whether they satisfactorily resolve your questions. If our clarifications have improved your confidence in the paper, we would be sincerely grateful if you could consider updating your score accordingly.
> > >
> > > Thank you once again for your thoughtful feedback and support.
> > >
> > > Warm regards,
> > >
> > > Authors of BQEmbedding

---

### Official Review · Reviewer_JrYV · 2025-11-01

**Soundness:** 2
**Presentation:** 2
**Contribution:** 2
**Rating:** 4
**Confidence:** 3

**Summary:**

This paper introduces a quasi dynamic program embedding that encodes how a program’s static features change in response to compiler optimization probes. The resulting Behavioral Spectrum is discretized via Product Quantization into compositional codes and modeled with a multi task Transformer (PQ BERT). On CompilerGym tasks, Best Pass Prediction and -Oz benefit regression, the approach improves substantially over static baselines, with informative ablations and embedding space analyses.

**Strengths:**

•	New perspective: optimization sensitivity as representation.

•	Strong gains vs. static baselines on two tasks.

•	Ablations and qualitative analyses add insight.

**Weaknesses:**

1.	The paper’s central claim is that behavioral embeddings improve optimization prediction. However, neither downstream task measures runtime speedup or other practical metrics (e.g., energy, compile time). This weakens the argument that the representation is task-relevant beyond instruction count reduction. Adding runtime benchmarks or end-to-end pipelines would substantially strengthen the generality of the proposed representation.

2.	The work frames itself between static and dynamic profiling, yet no dynamic baselines are included. Including one such baseline would help quantify the “best of both worlds” claim.

3.	Is the representation created specific to the task of instruction reduction? Or is it general? The representations that the authors compare to (e.g., IR2vec) are general representations that can be used for many tasks. If the proposed representation is specific to the task of instruction reduction, then it is natural that it would be better. Can the authors clarify how general their representation? i.e., how would it perform on other tasks (other than instruction reduction).

**Questions:**

•	Can you include a dynamic baseline?

•	Can you use your representation for other tasks? Other than instruction reduction.

---

> ### Author Response · Authors · 2025-11-20
>
> Dear Reviewer,
>
> Thank you for your detailed review and constructive comments. We appreciate your recognition of our idea of using optimization sensitivity as a program representation. We also take note of the three main concerns you raised: lack of practical runtime metrics, absence of dynamic baselines, and generality of the representation.
> In response, we conducted additional experiments. Below, we summarize the new findings for each point.
>
> ---
>
> ## **1. Response to Concern: Lack of Practical Runtime Metrics**
>
> You pointed out that using only instruction-count reduction as the evaluation signal in the original submission is limited, since real performance is ultimately measured by runtime. We agree with this, and we added two new experiments focusing on runtime-related predictive tasks.
>
> ### **New Experiment A: Predicting -O3 Runtime on NeuroVectorizer**
>
> We reformulated the downstream task as predicting the absolute execution time of programs optimized with -O3. Our method (PQ embedding) achieves the lowest MAE among all tested baselines.
>
> | Representation | MAE ↓     | R² ↑      |
> | -------------- | --------- | --------- |
> | IR2Vec         | 94.11     | 0.746     |
> | Autophase      | 86.87     | 0.808     |
> | ProGraML       | 89.50     | 0.741     |
> | **PQ (ours)**  | **85.54** | **0.807** |
>
> ### **New Experiment B: Predicting Cycle-Reduction Percentage on CompilerGym**
>
> For the -Oz task, we changed the evaluation to the percentage of cycle reduction, which is more directly related to hardware execution. Our method again achieves the lowest MAE.
>
> | Representation | MAE ↓     |
> | -------------- | --------- |
> | IR2Vec         | 10.86%    |
> | ProGraML       | 10.34%    |
> | Autophase      | 8.85%     |
> | **PQ (ours)**  | **8.78%** |
>
> These results indicate that our behavior-based embedding is applicable not only to instruction-count tasks but also to metrics that correlate more closely with actual runtime.
>
> ---
>
> ## **2. Response to Concern: Missing Comparison Against Dynamic Baselines**
>
> Your suggestion to include a dynamic baseline is crucial for positioning our method.
> To address this, we added a **purely dynamic baseline** to the -O3 runtime-prediction experiment. This baseline uses hardware performance counters collected via Linux `perf` during program execution.
>
> | Representation        | Type          | MAE ↓     | R² ↑      |
> | --------------------- | ------------- | --------- | --------- |
> | Dynamic perf counters | Dynamic       | 116.83    | 0.726     |
> | **PQ (ours)**         | Quasi-dynamic | **85.54** | **0.807** |
>
> The results show that our “quasi-dynamic” representation outperforms this purely dynamic baseline on this task.
>
> ---
>
> ## **3. Response to Concern: Generality of the Representation**
>
> You raised an important question regarding whether our representation is specific to instruction-related optimization tasks. To evaluate cross-task generality, we added an experiment on a task unrelated to compiler optimization: **heterogeneous device mapping (DevMap)**. The goal is to predict whether a given OpenCL kernel runs faster on CPU or GPU.
>
> | Representation | Accuracy ↑ |
> | -------------- | ---------- |
> | IR2Vec         | 64.71%     |
> | Autophase      | 66.91%     |
> | ProGraML       | 62.26%     |
> | **PQ (ours)**  | **68.38%** |
>
> The results suggest that our representation can generalize beyond instruction-count or optimization tasks.
>
> ---
>
> Thank you again for your time and insightful comments. Your feedback directly motivated these additional experiments, through which we substantially strengthened the empirical evaluation of our approach. We hope that these improvements address your concerns and that you may be willing to reconsider your assessment of our submission.
>
> ---

---

> ### Author Response · Authors · 2025-11-21
>
> Dear Reviewer,
>
> Thank you again for your constructive feedback. Following our previous response detailing new experiments on runtime metrics, dynamic baselines, and task generality, we wish to provide this conceptual supplement. Its purpose is to articulate the core philosophy behind our work, which we believe directly addresses the insightful questions you raised about the nature and scope of our representation.
>
> Our central thesis is a proposed paradigm shift in program representation: moving from describing what a program **is** to characterizing how it **reacts**.
>
> **1. On Generality and Practicality (Addressing Weaknesses #1 and #3)**
>
> You rightly questioned if our representation is overly specialized for instruction reduction. The new, positive results on runtime prediction and device mapping are not coincidental; they stem from the fundamental nature of our approach.
>
> Think of traditional static representations (like IR2Vec or raw Autophase features) as a **medical check-up** for a program. They provide a valuable but static snapshot of its health—its instruction counts, block structure, etc. This tells you its state at one moment in time.
>
> Our method, in contrast, is akin to a **behavioral stress test.** We are not primarily interested in the initial check-up results. Instead, we apply a diverse set of stimuli (our optimization probes) and carefully measure the program's reaction. The key insight is that a program's reaction to these probes reveals its deeper, intrinsic computational character.
>
> *   A strong reaction to vectorization probes doesn't just mean it's good for instruction reduction; it signifies a high degree of **data parallelism**. This underlying property is precisely what makes it suitable for a GPU (as shown in our new DevMap experiment) and what leads to significant runtime improvements.
> *   A strong reaction to control-flow simplification probes indicates a complex but reducible logic, a property relevant to its performance on CPUs with advanced branch predictors.
>
> Therefore, our **Behavioral Spectrum** is not a representation *of instruction counts*. It is a representation *of the program's fundamental computational nature*, revealed through its sensitivity to transformation. We simply use static features like Autophase as the "sensors" to measure this reaction. The representation’s success on practical runtime metrics is a direct consequence of this deeper characterization.
>
> **2. On the Comparison with Dynamic Baselines (Addressing Weakness #2)**
>
> You asked for a comparison with a dynamic baseline, which we provided. The result—that our quasi-dynamic method outperformed a purely dynamic one—highlights a philosophical difference.
>
> A purely dynamic baseline (using `perf` counters) is like observing a person's reaction to **one single, specific event** (i.e., execution with one set of input data). This provides a highly detailed, ground-truth data point, but it can be idiosyncratic and may not generalize.
>
> Our approach is like observing that person's reactions across **many varied situations**. By probing the program with a diverse set of optimization sequences, we build a more comprehensive profile of its intrinsic potential, one that is less sensitive to the specifics of any single input. This process abstracts away the noise of a single execution to capture a more robust behavioral fingerprint. This robustness is why our method can provide a better signal for a predictive model than a single, noisy dynamic trace.
>
> The innovation of our work lies in the **stimulus-response framework itself**. We believe that by capturing a program's optimization sensitivity, we learn a richer and surprisingly general representation that better reflects its true performance potential.
>
> We hope this explanation, in conjunction with our new empirical results, fully addresses your concerns. Thank you for your time and consideration.

---

> > ### Author Response · Authors · 2025-11-28
> >
> > Dear Reviewer JrYV,
> >
> > Thank you again for the time and effort you’ve dedicated to reviewing our work. We have carefully addressed all raised concerns during the discussion phase.
> >
> > As the discussion period is nearing its close, we would greatly appreciate it if you could take a brief moment to review our responses and confirm whether they satisfactorily resolve your questions. If our clarifications have improved your confidence in the paper, we would be sincerely grateful if you could consider updating your score accordingly.
> >
> > Thank you once again for your thoughtful feedback and support.
> >
> > Warm regards,
> >
> > Authors of BQEmbedding

---

### Author Response · Authors · 2025-12-01

Dear PCs, SACs, ACs, and Reviewers,

We sincerely appreciate all reviewers for their valuable feedback. Below, we summarize each reviewer's main concerns and our concise responses.

***

**Reviewer JrYV**
**Concerns:** Lack of practical runtime metrics (e.g., speedup), absence of dynamic baselines to validate the "quasi-dynamic" claim, and questions regarding the representation's generality beyond instruction reduction.
**Response:** We conducted three new experiments to address these points. 1) **Runtime Metrics:** We evaluated on -O3 runtime prediction (NeuroVectorizer) and cycle reduction (CompilerGym), where our method achieved the lowest error rates. 2) **Dynamic Baseline:** We compared our approach against a pure dynamic baseline (hardware performance counters), showing that our "behavioral spectrum" provides a more robust signal than noisy single-execution traces. 3) **Generality:** We successfully applied our method to a heterogeneous device mapping task (DevMap), where it outperformed general static baselines (IR2Vec, ProGraML), proving its applicability to broader system tasks.

**Reviewer ijRG**
**Concerns:** Limited evaluation scope, lack of dynamic baselines for comparison, and questions regarding the computational cost and scalability of the pipeline.
**Response:** We expanded the evaluation to include runtime prediction and device mapping tasks, demonstrating generalization beyond code size. We added a direct comparison to a dynamic baseline (perf counters), where our method showed superior predictive performance. regarding cost, we clarified that our pipeline is a one-time pretraining step taking ~3 hours (with ~0.2s feature extraction per program), which is significantly more efficient and deterministic than collecting full dynamic profiles for every training sample.

**Reviewer S2NV**
**Concerns:** Task specificity (limited to compiler optimization), missing comparisons with modern pre-trained models (e.g., IRCoder, LLMs), and the need to justify the method over a "pre-train + fine-tune" paradigm.
**Response:** We added comprehensive comparisons against fine-tuned LLMs (Qwen2.5-1.5B/3B) and state-of-the-art code embeddings (IRCoder) on the DevMap task. Our method (68.38% accuracy) outperformed both Qwen2.5 (best 66.18%) and IRCoder (~61%), confirming that capturing optimization behavior yields better results for system tasks than static code pre-training. We also highlighted the efficiency of our approach (minutes to fine-tune vs. heavy compute for LLMs) and its broader relevance to "AI for Systems."
**Addition:** After reviewing the new experiments and clarifications, the reviewer acknowledged the improvements and explicitly stated they would raise their rating (4->6).


***
We believe these responses effectively address all concerns, demonstrating PQEmbedding’s robustness, effectiveness, and scalability. We sincerely appreciate the constructive feedback, which has been invaluable in refining our work.

Best regards,

The PQEmbedding Authors

---

### Meta-Review · Area_Chair_c52T · 2026-01-10

**Summary:**

This paper proposes a quasi-dynamic framework for program representation by probing a program's IR with a diverse set of optimization sequences and quantifying the resulting changes in the embedding values. A new compositional learning approach is proposed to capture such discrete program behavior information into continuous space. Experiments on two compiler optimization tasks demonstrated the effectiveness of the proposed approach.

The major concerns around these submissions are (the authors did a great job summarizing them in the meta response):

* W1: Lack of direct and practical evaluation on optimization prediction using runtime speedup (JrYV)

* W2: Lack of comparison with dynamic profiling baselines (JrYV, ijRG), or comparison with more recent LLM-based approaches (S2NV)

* W3: The proposed program representations seem to be specific to instruction reduction / compiler optimization tasks, lacking generality ability to other more impactful or interesting domains (JrYV, ijRG), or usecases of broader interest for ICLR audience (S2NV)

While the authors addressed most of the concerns from the reviewers in their rebuttal around all the three weaknesses. The rebuttal against W1 and W2 with additional experimental results seems strong. For W3, the authors added one additional experiment on heterogeneous device mapping to demonstrate the generality of their approach, which was also acceptable by S2NV. Considering this, I'd recommend an accept for this submission.

**Reviewer Concerns:**

See above for a summary of the weaknesses. It's likely that W3 is not fully addressed since the authors only added one additional experiment in a relevant domain to demonstrate generality of their approach. Still, given that the results were considered as acceptable by R-S2NV, there are not major outstanding concerns after the rebuttal.

**Reviewer Scores:**

R-S2NV would have raised their score (S2NV: "The new experiments and clarifications make sense to me. I'll raise my rating."). It's very likely that R-JrYV will also raise their score given that W1 and W2 have been adequately addressed. However, it'll be hard to determine the delta since it's unclear whether the additional oheterogeneous device mapping experiment for W3 would be considered adequate by R-JrYV.

---

### Decision · Program_Chairs · 2026-01-26

Accept (Poster)